# Improved Eagle Strategy Algorithm for Dynamic Web Service Composition in the IoT: A Conceptual Approach

Venushini Rajendran [ID], R Kanesaraj Ramasamy *[ID] and Wan-Noorshahida Mohd-Isa [ID]

Faculty of Computing and Informatics, Multimedia University, Cyberjaya 63100, Malaysia;
venushinirajendran@gmail.com (V.R.); wan.noorshahida.isa@mmu.edu.my (W.-N.M.-I.)
* Correspondence: kanes87@gmail.com

**Abstract:** The Internet of Things (IoT) is now expanding and becoming more popular in most industries, which leads to vast growth in cloud computing. The architecture of IoT is integrated with cloud computing through web services. Recently, Dynamic Web Service Composition (DWSC) has been implemented to fulfill the IoT and business processes. In recent years, the number of cloud services has multiplied, resulting in cloud services providing similar services with similar functionality but varying in Quality of Services (QoS), for instance, on the response time of web services; however, existing methods are insufficient in solving large-scale repository issues. Bio-inspired algorithm methods have shown better performance in solving the large-scale service composition problems, unlike deterministic algorithms, which are restricted. Thus, an improved eagle strategy algorithm method is proposed to increase the performance that directly indicates an improvement in computation time in large-scale DWSC in a cloud-based platform and on both functional and non-functional attributes of services. By proposing the improved bio-inspired method, the computation time can be improved, especially in a large-scale repository of IoT.

**Keywords:** bio-inspired algorithm; Internet of Things; improved eagle strategy; large-scale repository; web service composition

## 1. Introduction

In this emerging age of technology, the Internet of Things (IoT) has a large role in the future of smart cities and is expected to bring a significant amount of cash flowing through the market in the upcoming years. IoT is expected to be found in many fields such as smart business, building, and home automation, healthcare, and the automotive industry [1]. IoT relies on Service-Oriented Architecture (SOA), a basic independent component for protocol communication over the Internet or a network (such as HTTP), and Web Services (WS), a standard of SOA for implementing dynamic and supplying business processes.

WS can act as a connection between IoT devices and data storage. WS are used as a medium to process data and are required for storing the processed data into the respective database. WS are now being used by several industries to share information for functions across platforms [2]. Instead of invoking several WS to complete the business processes for different functional requirements such as 'registration', 'login', 'sending email', 'chat', etc., Web Service Composition (WSC) was developed. WSC is the coordination or combination of several WS into a single web service. WSC will be the intermediate medium to process and store data dynamically; thus, research must consider several types of composition methods.

Of late, integrating WSC into the IoT devices is being perceived as an emerging paradigm also known as the IoT Service Composition (IoTSC). With the idea of an IoT device serving as a cloud database, it can be inferred that IoTSC may help save resources and costs; however, overheating of IoT devices is a major issue because it can lead to device failure. One of the key reasons for overheating is that so many processes are performed by the IoT devices, 24 h, 7 days a week. Furthermore, according to [3], most approaches do

not satisfy certain required criteria and ignore the fundamental aspect of IoT, especially on the real-time constraint, monitoring, and fault tolerance. This is made worse due to limited evidence on the lack of monitoring on temperature, energy [4], and power consumed that overheat the devices. Thus, in this research, several types of composition methods are considered to tackle this major issue.

As a result of the high computation time in selecting WS, the time taken to complete the WSC had increased, especially in a large-scale repository. As mentioned by [5], the number of services had been increasing, but the existing solutions were insufficient to solve the large-scale problems. Thus, we can summarize that poor performance during service selection in large-scale service repositories and selecting the correct service to avoid user dissatisfaction were among the main issues. We hereby put this research forward with the following objectives:

1.　To reduce computation time in selecting WS;

To propose a hybrid model for IoT devices to reduce the execution time on selecting the service from a large-scale cloud-based web service repository for DWSC. This model to be proposed is to increase the performance, which directly indicates the reduction in computational time on large-scale DWSC. This proposed model can be implemented into IoT devices. Hence, the process and overheating issue may be reduced.

2.　To select the correct service in selecting WS.

To select the correct service based on user requirements for DWSC. This may be able to improve user satisfaction and maintain the SLA.

## 2. Literature Review

According to [3,6], the challenges of performing service composition in IoT were heterogeneity and the rapid increase in the number of devices and services [7], including the costs of service composition in IoT, which were yet to be solved. Additionally, other issues raised by the researchers were the monitoring mechanisms of service composition that should be implemented, in which the consideration of most researchers was only on the availability and unavailability of the services or other Quality of Services (QoS) attributes. According to the authors, the motivations for the adaption of service could be summarized in two points, one to ensure continuity of services to the user, and two, to improve the user interaction with the environment. There were some comparative studies conducted by researchers, but due to lack of implementation, there were no results to prove the suitable approach to solve the challenges.

As mentioned by [8], the aim of their research was to present the comparative investigation of the selected service composition techniques in the IoT. They presented comparisons between the similarities and alterations among the current service composition in the IoT based on appropriate parameters. The authors also determined the important areas where future research could expand the service composition techniques in the IoT by offering the comparison and briefing techniques; however, the author claimed that most of the articles tried to improve execution time, scalability, and cost, but availability and reliability were rarely considered.

Based on the finding by [9], a traditional optimization technique was applied to solve the selection and composition of cloud services; however, it still suffered from slow convergence speed, a large number of calculations, and falling into local optimum. The author also reported that user dissatisfaction or violation of the Service Level Agreement (SLA) was a major problem. Thus, the author proposed a hybrid optimization method with the combination of Particle Swarm Optimization (PSO) and the Fruit Fly Optimization Algorithm (FOA) to improve the performance in terms of fitness value, execution time, and error rate; however, the proposed methods had a drawback on computation time for large-scale implementation.

As explained by [10], the number of web services is increasing dramatically, which results in difficulties for users to find the desired web services. The author claimed that

most of the existing methods recommend services according to accurate prediction on rating or QoS values. Thus, the author proposed a model named Support Vector Machine (SVM)-based Collaborative Filtering (CF) Service Recommendation Approach (SVMCF4SR) to correctly rank the services rather than to accurately predict based on historical data—in this case, good QoS value services will be ignored depending on user ratings.

Ref. [11] stated that the QoS-aware cloud service composition problem is also known as the NP-hard problem, and many methods are studied to solve this issue and yet cannot solve it due to the large scale and growing repository. A Hybrid Genetic Algorithm (HGA) method was proposed to solve the composition problem. The proposed HGA method is the combination of a Genetic Algorithm (GA) and FOA. GA is a method widely used to solve several optimization problems, whereas FOA was inspired by the food-finding behavior of fruit flies. FOA was employed for local search to maintain the balance between exploration and exploitation. In this case, the scenario is yet to be tested in the distributed cloud environment.

Distance plays an important role in the K-means algorithm to find the point-to-point distance between web services. According to [12], Manhattan distance achieves good results. In the Manhattan distance function, the distance between two points is the sum of the absolute differences of their Cartesian coordinates. For the most part, it is the difference between the x-coordinates and the y-coordinates added together; thus, the Manhattan distance $d(x,y)$ can be defined as below [12,13]. The algorithm of the Manhattan distance is as follows:

$$d(x,y) = \sum_{i=1}^{k} |x_1 - y_1|$$

A physical network of objects (or "things") that have been implanted with sensors, software, and other technologies to link and share information between devices and systems over the Internet is referred to as the Internet of Things (IoT). Physical objects can be empowered to create, receive, and seamlessly share data via the Internet of Things. Home automation, wearable gadgets, and personal health are examples of IoT applications at the consumer level. Retail, industrial IoT, smart utilities, and health care are all examples of IoT applications at the business level [14].

## 3. Research Methodology

We carried out this research by proposing a new method to solve the problem of improper service selection of service composition in a large-scale WS. Figure 1 shows the activities that were carried out throughout this research, which involved four different phases A (Identify Problems/Research Gaps), B (Modelling), C (Development), and finally D (Testing).

### 3.1. Identify Problems/Research Gaps (Phase A)

In this first phase, we reviewed the existing works on the domain-specific requirement of WSC selection techniques, IoT, and large-scale databases. Based on the existing work on service selection methods, we identified those existing approaches incapable of giving a good performance, especially for a large-scale repository that can lead to high computation time in dynamic WSC.

We also reviewed methods in selecting the functional attributes of correct services since these were often overlooked by the researcher. Most of the proposed approaches only focus on Quality of Service (QoS), which is a non-functional attribute of services such as the response time. Focusing only on the QoS attributes can cause incompatibility of selected services during service composition. This can also violate user requirements and reduce customer satisfaction.

**A) Identify Problems / Research Gaps**

- Identify the problems in large-scale DWSC on cloud-based platform
- Analyze the existing method in DWSC
- Study the possible problem occurs and implementation domain

**B) Modeling**

- To model the web service discovery method to reduce the execution time in large scale repository
- To model the web service selection method to improve the response time in dynamic service selection

**C) Development**

- Development of prototype

**D) Testing**

- Testing and Validation

**Figure 1.** Flow of the activity.

### 3.2. Modeling (Phase B)

The next phase was the modeling phase. In this phase, simulations of existing methods such as the Eagle Strategy (ES), Whale Optimization Algorithm (WOA), and Particle Swarm Optimization (PSO) were carried out by using the Netlogo simulation tool with the same number of datasets and environments. Netlogo is a multi-agent programmable modeling environment. The ES method gave the better performance compared to the other methods, and researcher [15] mentioned the same, in which ES outperformed others compared to the standard algorithms such as Genetic Algorithm (GA), WOA, Discrete Guided Artificial Bee Colony (DGABC), Hybrid Genetic Algorithm (HGA), and Greedy Randomized Adaptive Search Procedure (GRASP).

Based on Figures 2 and 3, we employed 500 food items to replicate WOA and PSO using a simulation. Figure 4 depicts the results of the same simulation conducted for ES. The results suggest that ES outperformed WOA and PSO in terms of performance (time). As a result, we selected ES and proceeded with the simulation using a reduced space, as illustrated in Figure 5.

Similar good performance can be seen from our simulation in Figures 3–5. In Figures 3 and 4, the ES method was seen to work better in terms of performance on selecting the services when the search space was small. The advantage of ES was utilized fully in this research, where ES was able to select the services in less response time (indicating better performance) on a small-scaled repository. Additionally, existing ES approaches only focused on non-functional attributes of services, which is the QoS. Our proposed method focused on both functional and non-functional attributes of services as well as focusing on the response time on a large-scale repository. Thus, we named our proposed method the Improved Eagle Strategy (IES).

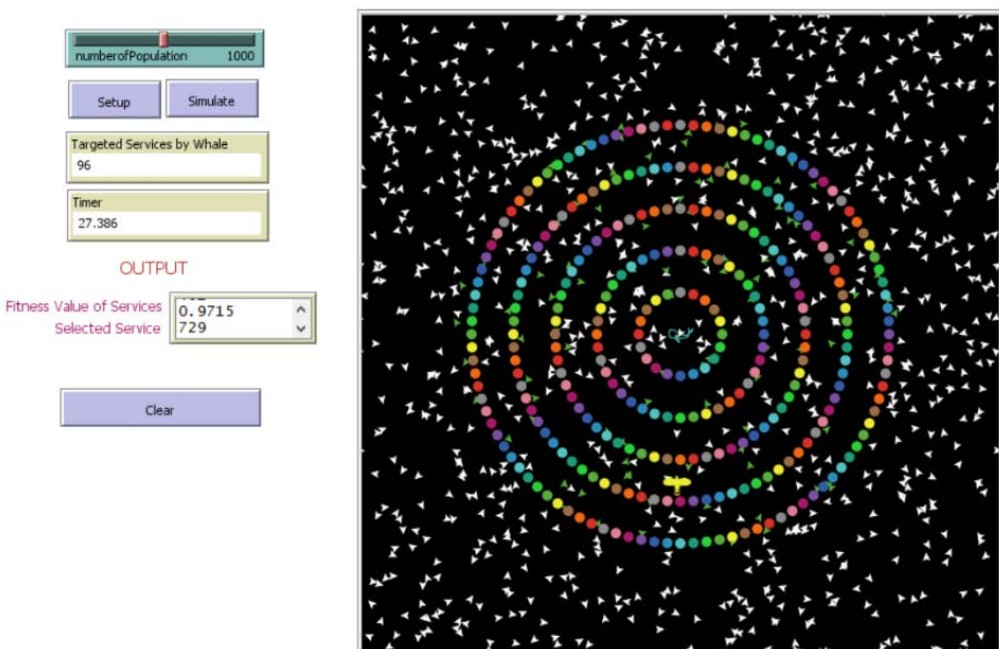

**Figure 2.** Simulation of the whale optimization algorithm from beginning at time = 0 with the number of food = 500, until the end, the number of food = 1.

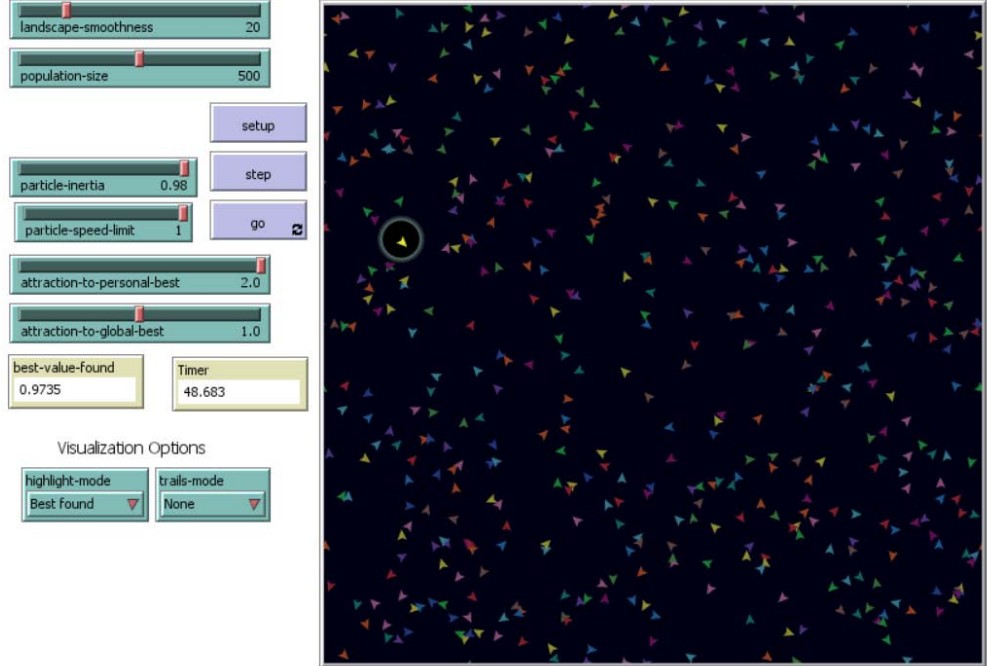

**Figure 3.** Simulation of particle swarm optimization from beginning at time = 0 with the number of food = 500, until the end, the number of food = 1.

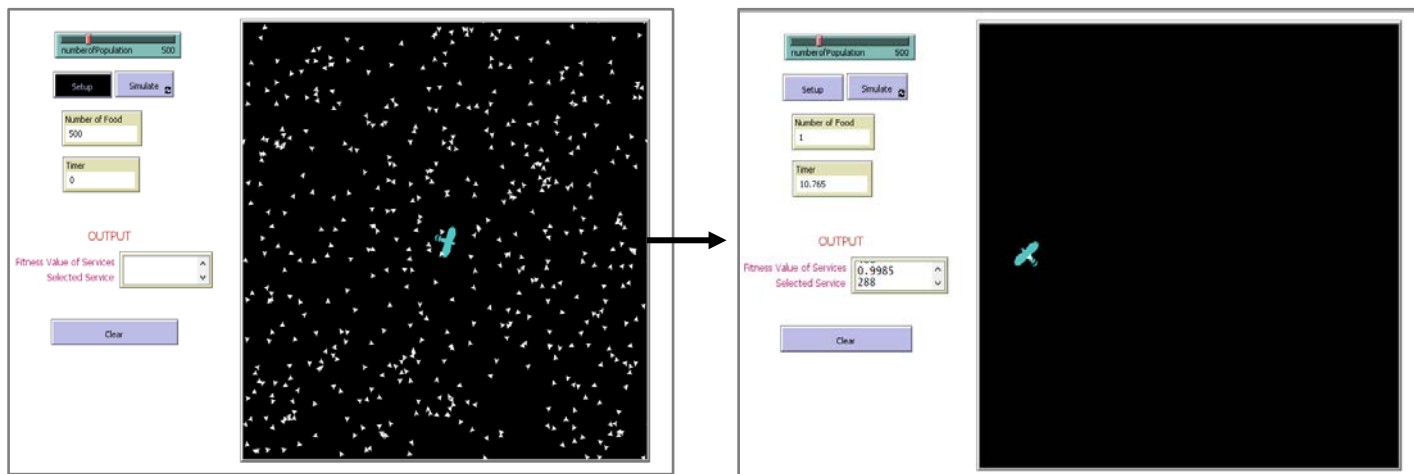

**Figure 4.** Simulation of eagle strategy by ESNormal from beginning at time = 0 with the number of food = 500, until the end, the number of food = 1.

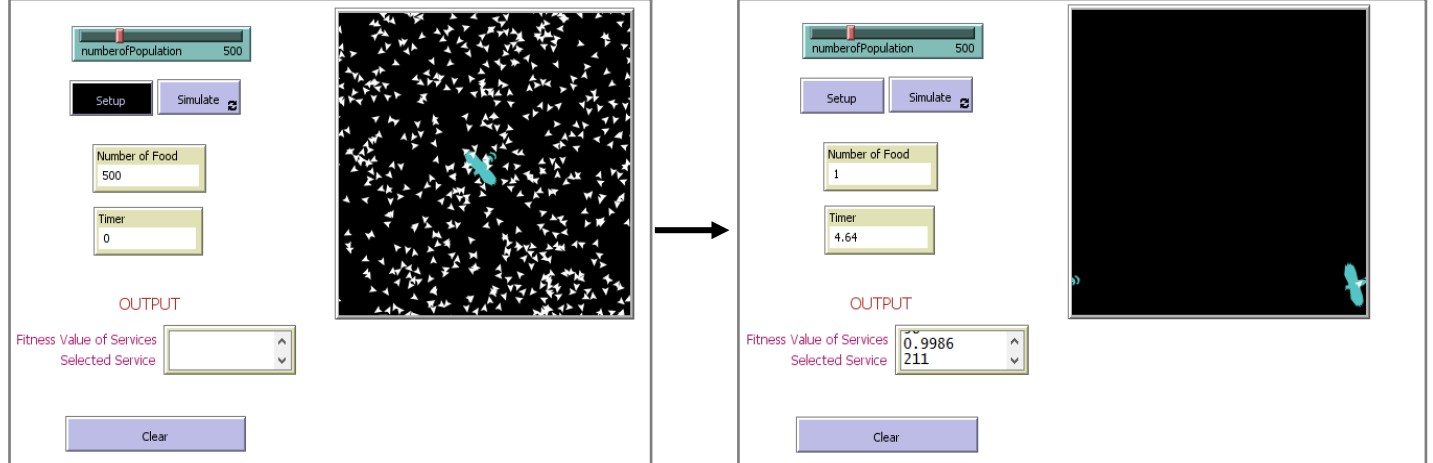

**Figure 5.** Simulation of eagle strategy by ESReduced from beginning at time = 0 with the number of food = 500, until the end, the number of food = 1.

Figure 5 shows the graph of the time taken to complete the iteration by standard ES for 1000 services. The time taken to complete the iteration was less than when the search space was small. ESNormal indicated the standard search space by Netlogo simulation; meanwhile, ESReduced indicated the search space was reduced at 50% from the standard search space to evaluate the efficiency of the ES algorithm on selecting the services, especially in a large-scale repository.

It can be seen in Figure 4 that the time taken for ESNormal had been increasing somewhat linearly with the increase in the number of services compared to ESReduced, which was somewhat constant; therefore, our proposed method had selected services from the large-scale repository where the services would be clustered into small groups based on the functional attributes of the services. These attributes were based on the method names that were retrieved from the Web Service Description Language (WSDL).

### 3.3. Development (Phase C)

In this development phase, there were several activities conducted, as shown in Figure 5.

We made use of the WS-DREAM repository [16], which is an open dataset that hosted the services from different countries and was collected from real-world WS. As this dataset had been widely used in research, thus, the validation would be accurate for comparing the outcome. In this research, we focused on the development of WSC in a distributed

cloud environment. We used Microsoft Azure as a cloud computing service for building, testing, deploying, and managing applications and services through Microsoft-managed data centers. In the next activity, we calculated the semantic similarities of services. For each service, the method name and input–output parameters were extracted, and semantic similarities were calculated by using the Wordnet 3.0 algorithm.

The third activity in this phase was clustering the services based on their method name. To make use of the advantage of the traditional ES, the clustering method was applied to reduce the search space to ease the service selection process; therefore, the proposed method was able to select the service in less computation time. The K-means clustering technique was applied in our proposed method [17] by using the Microsoft Azure Machine Learning Studio. This text clustering technique was accurate and resulted in a better accuracy and F-measure based on the findings of [18]. The fourth activity was the implementation of our proposed algorithm as a web service by using the Microsoft Visual Studio. The final activity of the development phase was a development of a Windows workflow composition by using the Windows Communication Foundation (WCF) for 24 Smart Toilet microcontrollers.

### 3.4. Testing (Phase D)

In this testing phase, there were three main objectives to be fulfilled. It began with evaluating the execution time of our proposed method on selecting the services from a large-scale repository. Existing methods from other authors and our proposed method were simulated by using the simulation tool Netlogo. Retrieval of the execution time on selecting the services was tested on the same environment and dataset. The results were compared by using a data visualization method.

The second objective was to evaluate the correctness of the selected service of the proposed method. Figure 6 shows a flowchart detailing how the test was carried out. The functional requirement was tested in this phase, where the proposed method would collect the parameter from the end-user, which was the method name. To test the correctness of the service, the selected services were intentionally removed from the repository, and the outputs were re-validated.

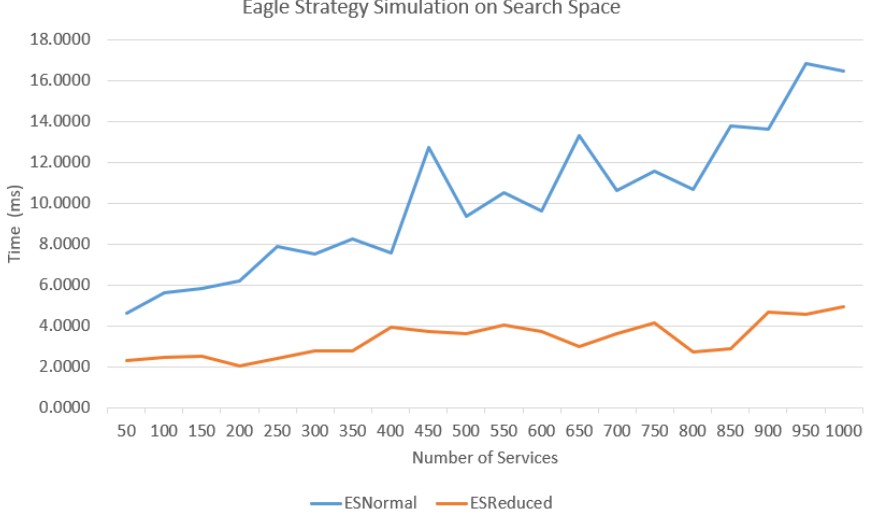

**Figure 6.** Graphical view for the time taken to complete the iteration.

The dynamic selection was tested in this phase as well by using a substitution method [19,20], as shown in Figure 7. According to [20], the substitution was one of the self-healing techniques, which was capable of diagnosing the failure and applying a suitable repair in the composition of WS. This had ensured the composition to be able to service the end-user in DWSC, as illustrated in Figure 7; Algorithm 1 describes how the substitution worked.

**Algorithm 1.** Substitution of Services.

| | |
|---|---|
| 1: | Begin |
| 2: | Monitor the service |
| 3: |       Begin If (availability = true) |
| 4: | Continue the web service workflow |
| 5: |       Else |
| 6: | //apply improved eagle strategy method |
| 7: |       End If |
| 8: | End |

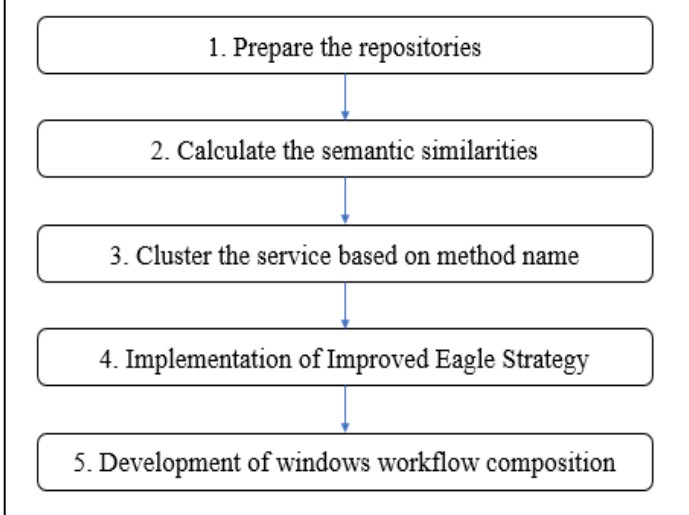

**Figure 7.** Activities carried out during the development phase.

## 4. Improved Eagle Strategy

In this section, the proposed method with the combination of ES and K-means clustering technique is discussed first, followed by the presentation of the algorithm. We named the proposed method the Improved Eagle Strategy (IES) algorithm, and the overview is illustrated in Figure 8. The proposed method is an improved version of the ES method that may be able to obtain balance exploration and exploitation and may also outperform other existing bio-inspired methods. ES was developed by Yang and Deb [21]. This technique has been used by researchers to enhance the efficiency of metaheuristic algorithms. Figure 9 below overview of IES.

We propose an improvement to the ES algorithm by applying the K-means clustering technique to cluster similar WS, since WS are growing in numbers, and this growth makes retrieval of the correct services upon user request difficult. To recognize the necessary WS, we propose to make it compulsory to cluster the WS to make the services easily discoverable. The WS cluster would arrange and manage the services to narrow down the scope of the search space.

K-means is a common clustering technique used in text clustering and was introduced by MacQueen James and other researchers in 1967 [18]. The algorithm initially assigns a random initial cluster centroid. This method is also widely used and performs better for big datasets [12]. This algorithm can use the maximum similarity measure to assign each service for a centroid based on a similar cluster. The algorithm of the K-means clustering technique is presented in Algorithm 2.

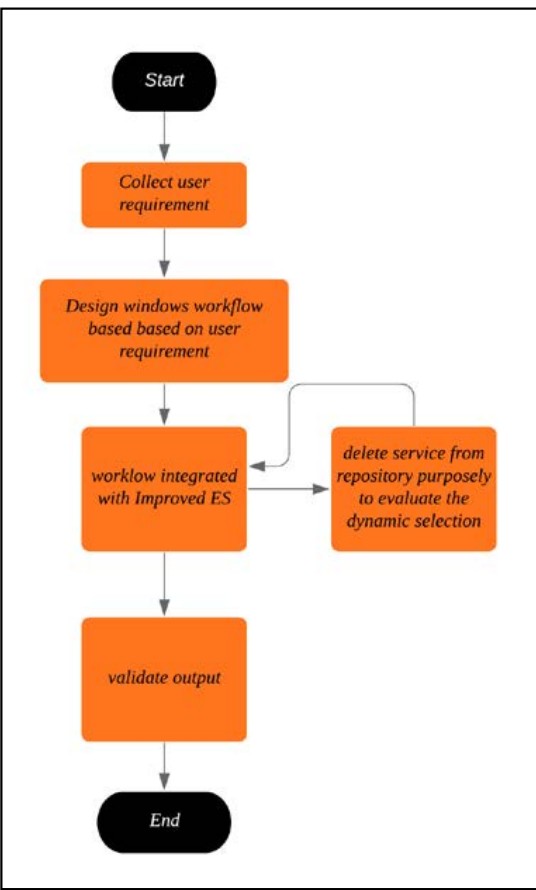

**Figure 8.** Steps to test the correctness of the selected service.

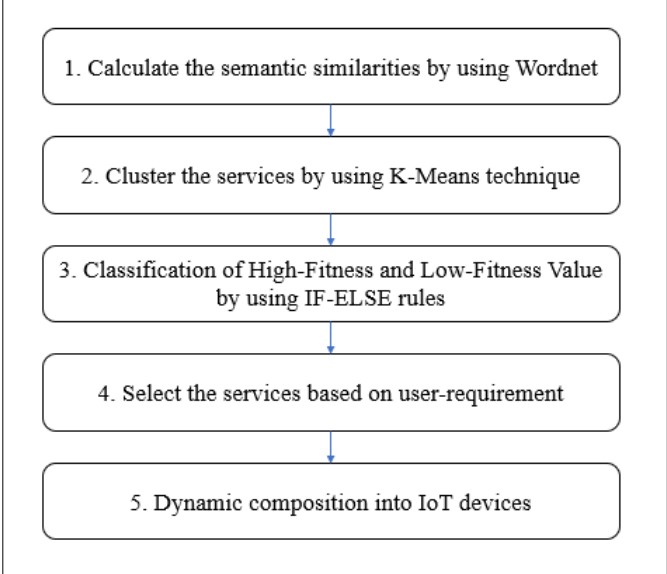

**Figure 9.** Overview of Improved Eagle Strategy (IES).

---

**Algorithm 2.** K-means clustering [18].

---

1:  Input: A collection of services, K is the number of all clusters, D is the set of services
2:  Output: Assign D to K.
3:  Termination criteria
4:  Randomly choosing K services as clusters centroid C = (c1, c2,..., cK).
5:  For all d in D do
6:  If (Pe < rand), do
7:      let j = argmaxk?{1toK}, using the cosine similarity
8:      Assign d1 to the cluster j, A[i][j] = 1
9:      Update the cluster centroid using Eq(1)
10: End For

---

In this research, the K-means clustering technique was applied to the services to group the services within similar categories. Before applying the clustering technique, semantic similarities were measured by using WordNet 3.0, which is a free and publicly available software for download. Further, the structure of WordNet 3.0 makes it a useful tool for computational linguistics and natural language processing. Before calculating semantic similarities, the input was retrieved from WSDL and used to extract important data, such as the method name and input–output parameters. The method name was the functionality of the service; the input and output elements specify the abstract message format for the solicited request and response, respectively.

Semantic similarities calculation is an important phase because the clustering of WS requires the calculation of distance between services. WS clustering consists of extracting characteristics of WS from the repository and discovering the similarities between the clusters or groups. Once the services have been clustered, the fitness value [22,23] for each service was calculated using Equation (1) and classified as having a high or low fitness value by comparing each service with a threshold value [20]. Equation (2) is the formula used to calculate the threshold value of each service. Once both fitness value and the threshold value are identified, Equation (3) was used to identify if the service owns a high fitness value or low fitness value. The pseudocode of ES is presented in Algorithm 3.

---

**Algorithm 3.** Eagle Strategy.

---

1:  Objective function f(x)
2:  Initialization of sample space
3:  While (t < maximumnumberofiterations)
4:      Do Global Exploration by randomization
5:        Fitness Evaluation and finding a promising solution
6:          If (Pe < rand), do
7:            Local exploitation by efficient local optimizer
8:              If (a better solution is found)
9:                current best solution is updated
10:             End
11:         End
12: t = t + 1;
13: END

---

$$\beta = \left( \frac{\alpha1}{\alpha1_{max}} \right)(a\%) + \left( \frac{\alpha2}{\alpha2_{max}} \right)(b\%) + \left( \frac{\alpha3}{\alpha3_{max}} \right)(c\%) \tag{1}$$

$\beta$ = FitnessValue
$\lambda$ = Quality of Service (QoS) attribute value
$a$, $b$, $c$ = Weightage based on percentage

$$ThresholdVal = \frac{sum\ of\ fitness\ value\ in\ the\ repository}{Total\ number\ of\ services\ in\ repository} \tag{2}$$

$$
\begin{aligned}
&If, \\
&FitnessVal \ \geq ThresholdVal = HighFitnessZone \\
&Else, \\
&FitnessVal < ThresholdVal = LowFitnessVal
\end{aligned}
\tag{3}
$$

For selection purposes, high fitness value services are first prioritized. Selecting services that focus on both functional and non-functional attributes [24] is the advantage of our proposed IES. Algorithm 4 shows the algorithm of our proposed method that is initiated by obtaining the requirements from the end-user. These are the functional attributes followed by the non-functional attributes requirement.

---

**Algorithm 4.** Improved Eagle Strategy.

---

1:  Start;
2:  UR = Request for User Requirement (Searching Term);
3:  Lowest_Value_Cluster_Simmilarity = 0;
4:  Chosen_cluster;
5:  while (search for cluster in repository) {
6:      service = Select Service Randomly;
7:      Similarity_point = (UR, service);
8:      if (Similarity_point < Lowest_Value_Cluster_Simmilarity)
9:          Lowest_Value_Cluster_Similarity = Similarity_point
10:         Chosen_cluster = cluster
11:         Cluster_Identified = Chosen_cluster;
12:         low_similarity Service_Point = 0;
13:             while (service from each Cluster_Identified sort by fitness value ascending) {
14:                 Similarity_point = (UR, service)
15:                     if (Similarity_point < low_ similarity Service_Point)
16: Chosen_service = service
17:                     low_similarity_Service_Point = Similarity_point
18:                 }
19: }
20: Compose Service(chosen_service);
21: End

---

By focusing on the functional attributes, we are able to reduce the incorrect service selection and reduce the failure of service composition that violates the SLA [25]. Thus, selecting the correct service for composition based on the user requirement may improve user satisfaction [26]. The services from the repositories are clustered based on the method name, which would be retrieved from the WSDL; therefore, it is selected when the user searches based on the keyword.

At the same time, our proposed method is focused on the non-functional attributes of the QoS that may improve the efficiency of the WSC in terms of availability, response time, and performance, which are the computation time of the services [22,27]. The services are selected for composition based on the user requirement to satisfy the functional requirement that is based a specific cluster. Then, within the same cluster, a service is selected based on the non-functional attributes. The IES algorithm is implemented as a WS that is integrated with IoT devices to support the consideration of real-time data. Additionally, it may support instant information where IoT devices are retrieving a lot of data; therefore, our proposed algorithm may fulfill the need for a robust web service solution to handle the complexity in real time.

## 5. Analysis

Based on the simulation findings, our proposed approach IES outperformed existing approaches in terms of performance. To observe service selection, we performed the simulation with 1000 services in a dataset. According to the results of an experiment conducted by [8], PSO failed after it reached 800 services in its dataset. When compared to

WOA and PSO, ES outperforms both. Since it performed better than the others, we chose to modify the ES with K-means to address the issue in dynamic web service selection in a large-scale repository, which was one of the study's objectives.

According to [28], numerous researchers have developed combinatory or hybrid approaches to attain the least time possible; however, previous methodologies are inadequately efficient in terms of compiling the important services in a reasonable amount of time due to the ever-increasing number of services, which leads to the growth of the search space [5,28].

Aside from that, while selecting the proper service for service composition from a large-scale repository without human interaction, the functional attribute must be taken into account. Nonetheless, the majority of proposed methodologies for service selection focus solely on non-functional attributes of service effect to reduce SLA violations [29,30].

When the scope of the search is limited, the proposed approach can select the service in the shortest amount of time, hence saving time. The lower scale was achieved by the application of the K-means clustering technique. Aside from that, when selecting a service, the suggested strategy takes into account both functional and non-functional attributes. As a result, user satisfaction is achieved, which is related to the second purpose of our research.

## 6. Conclusions

In the IoT, WS are crucial components. The physical world was directly tied to IoT service composition, and DWSC became a key difficulty because it was performed in real time and in response to user requests. The IES method was designed to meet the user's needs, and it met both functional and non-functional IoT requirements. The comparison of WOA, PSO, and ES was made in the same environment and with the same dataset. When compared to WOA and PSO, it has been proven that ES outperforms both. Furthermore, we were able to achieve a balance between exploration and exploitation using this strategy, and we were able to overcome the issues of high reaction time in expanding repositories. The issue of service selection is resolved by comparing the fitness value of each service. The fitness value is derived from the response time and availability of the services. The study focuses on IoT resources exposed via RESTful services in a distributed cloud environment. The quality of such services and their performance play an essential part in business processes. The paper outlines a novel approach to improving IoT system performance through the use of RESTful services and correct service selection. As a result, this repository can accommodate a wide range of services. This is one of the study directions that grew out of Mike Papazoglou's web service research roadmap [14], which was also followed by [31]. The purpose of this study was to demonstrate how the web service composition component of the second component of the web service research roadmap works. For the composite service to fulfill functions such as coordination, monitoring, conformance, and Quality of Service (QoS), service aggregators develop specifications and/or code for the composite service. According to the proposed roadmap, the focus of this article is on QoS.

**Author Contributions:** V.R. was involved in the simulation's modelling and execution. R.K.R., who is also the corresponding author, assisted in verifying the model and editing the manuscript. W.-N.M.-I. assisted in the editing of the manuscript. All authors have read and agreed to the published version of the manuscript.

**Funding:** This research was funded by TM R&D, grant number MMUE/180023" and "The APC was funded by MULTIMEDIA UNIVERSITY.

**Conflicts of Interest:** The authors declare no conflict of interest.

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
