# Peer review of "Improved Eagle Strategy Algorithm for Dynamic Web Service Composition in the IoT: A Conceptual Approach"

_futureinternet, doi:10.3390/fi14020056_

Round 1

Reviewer 1 Report

The authors propose an improved version of the Eagle-Strategy (ES) algorithm for Dynamic Web Service Composition (DWSC) in IoT. Overall, this paper is easy to follow. This reviewer has the following comments.

1) My main concern is that this paper lacks experiments to quantify their system performance. Without experiment results, this reviewer cannot understand the effectiveness of the improved ES (ES + K-means) compared to the standard ES in reducing the searching time. Also, the tradeoff between the time reduction and the accuracy degradation is unclear. 

2) This paper is targeting IoT scenarios. However, the connection between this paper and IoT is weak. The authors need to clarify the challenges brought by IoT that make its DWSC for IoT unique and thus existing DWSC works cannot be directly applied. Simply mentioning IoT devices suffer from over-heating is not enough.

3) In Section 2.2, the authors mention three algorithms: ES, WOA, and PSO. Since the authors have conducted experiments with these algorithms, it is better to present the results of their experiments, rather than directly jumping to the conclusion that ES is the best without any strong evidence. Merely citing exiting work without own experiment results to justify that ES is the best makes this reviewer doubt whether the authors conduct proper experiments. In addition, the settings of the experiment need to be explained (what dataset? How to conduct experiments?)

4) Please pay more attention to references. Some references lack the publication source (e.g., [3]), some journals lack the volume/issue number/page numbers, etc. 

Author Response

Thank you for the reviews and comments. We have made the changes accordingly. 

Reviewer 2 Report

Authors have proposed an Improved Eagle Strategy method to solve the service composition problem.

  1. Related work section is missing - you need to clearly describe most important relevant works and what is the contribution of your new method
  2. Your work with WSDL, but today the most of the services are RESTfull services, so please elaborate how your approach can be used with REST services
  3. Please elaborate on performance of your approach and compare it with similar related approaches from the literature - How is your approach better than similar ones
  4. Conclusion section is too short and it does not shows the scientific contribution of your study

Author Response

Thank you for your comments and suggestions. We have made the changes accordingly. 

Reviewer 3 Report

This article proposes an improved eagle-strategy algorithm to increase the performance of web services composition, thus improving the computation time in large-scale DWSC in a cloud-based platform and on both functional and non-functional attributes of services.

In general, the work is interesting and has potential. Yet, the following major concerns need to be addressed:

  • The references are mostly old. The authors need to discuss more recent literature.
  • There’s a need to discuss the proposed solution against a large literature proposing trust-based web service composition such as:
    • "A survey on trust and reputation models for Web services: Single, composite, and communities." Decision Support Systems74 (2015): 121-134.
    • "Integrating trust with user preference for effective web service composition." IEEE Transactions on Services Computing10, no. 4 (2015): 574-588.
    • "Integrating trust with user preference for effective web service composition." IEEE Transactions on Services Computing10, no. 4 (2015): 574-588.
  • Line 7 in Algorithm 2 contains some errors. It needs to be reshaped.
  • More simulations and comparisons need to be added to make the work more interesting.

Author Response

(The authors gave the same response as above.)

Round 2

Reviewer 1 Report

The authors have answered my questions, and I am OK with the paper.

Author Response

Good day,

 Thank you for your reviews. 

We have updated the manuscript and submitted the revised manuscript. 

All the required changes have been included. 

Reviewer 3 Report

The authors adequately answered my comments.

Author Response

(The authors gave the same response as above.)
